# Validation and evaluation of a tablet-based dietary record app for adults aged 70 and above

**Jette Hinrichsen**\*, **Vincent Quinten, Rebecca Diekmann**

Junior Research Group Nutrition and Physical Function in Older Adults, Department of Health Services Research, Carl von Ossietzky University Oldenburg, Germany

\* Jette.Hinrichsen@uol.de

## Abstract

### Objective

This study aimed at validating the dietary recording functionality of the NuMob-e-App, developed for adults aged 70 and above, against the 24-hour dietary recall reference standard.

### Methods

104 independently living adults (mean age 75.8±4.1 years; 58% female) from northwest Germany participated. They recorded their dietary intake on three consecutive days using the App. In parallel, we conducted a structured 24-hour dietary recall via telephone. Nutritional intake was analysed for energy, macronutrients, and food groups defined by the German Nutrition Society. Data were analysed for equivalence using Two One-Sided Tests (TOST), agreement using Intraclass Correlation Coefficients (ICC), and systematic differences using Bland-Altman plots.

### Results

Equivalence could be shown in 20 of the 44 compared variables, ICC varied between 0.677 to 0.951 for the four macronutrients and between 0.714 and 0.968 for the seven food groups. The Bland-Altman plots showed tendency to underestimation by the app in most variables and relatively narrow limits of agreement.

### Conclusions

The NuMob-e-App demonstrated good relative validity for assessing energy, carbohydrate, and protein intake, as well as selected food groups in older adults. While equivalence was not achieved across all 44 variables, agreement was particularly strong for protein and beverages. A general tendency toward intake underestimation by the app was observed. These findings support the app's potential for use in preventive dietary self-monitoring among seniors.

**Data availability statement:** In the informed consent form that all participants signed, it is explicitly stated that their data will not be shared with third parties. Therefore, the data from this study cannot be made publicly available in order to comply with the approved consent and data protection regulations. However, I am happy to make de-identified data available upon reasonable request to qualified researchers, in accordance with ethical and legal requirements. The study was approved by the Medical Ethics Committee of the Carl von Ossietzky University Oldenburg (Ethics vote number: 2023-172) and registered under DRKS-ID: DRKS00032600. Data access requests may be directed to uac.medizin@uol.de. In order not to violate the agreement with our participants, we are unable to make the data publicly available - we thank you for your understanding!

**Funding:** R.D. and V.Q. are supported by the Federal Ministry of Research, Technology and Space (BMBF) (Grant 01GY2102).

**Competing interests:** The authors have declared that no competing interests exist.

## Introduction

With demographic aging and rising health risks in older populations, malnutrition has become a central concern. It is not only linked to acute and chronic illness, but also to reduced physical resilience and longer rehabilitation, and increases the risk for frailty, sarcopenia, and hospitalization [1]. Older adults are particularly vulnerable due to physiological changes, reduced appetite, and external factors such as social isolation or limited mobility [2,3]. Despite their lower energy requirements, their need for nutrients such as vitamins, proteins, and fibre often remains the same or even increases with age, creating a need for a nutrient-dense diet, which may require adjustment [4,5]. In this context, 22.8% of older adults in Germany were malnourished in 2010, and nearly half of them (46.2%) were at risk [6].

Early detection and management of nutritional deficiencies in this setting is essential not only for maintaining health, quality of life, and independence, but also for preventing the economic burden associated with malnutrition-related hospitalizations and long-term care [3,7]. Accordingly, early dietary intervention is critical in an aging population with increasing chronic disease prevalence [1,2].

Dietary assessment and regular feedback on one's diet are key tools in diagnosing malnutrition and supporting dietary counselling, not only as a preventive measure, but also as a follow-up to rehabilitation [5,8].

Among the most common methods are food records and 24-hour-recalls, both of which provide individual dietary data [9]. Nevertheless, collecting and evaluating such data is very time-consuming and therefore also associated with considerable costs [7]. In times of shortage of healthcare professionals [10], it is important to use the available resources more effectively.

To address this shortage, empowering older people to monitor and manage their nutrition independently at home is crucial. This not only promotes well-being and satisfaction, but can also prevent or delay care facility admission. To date, more than 95% of seniors in Germany live in their domestic environment [11].

Maintaining the nutritional status through easily accessible tools such as digital nutrition apps can therefore help prolong independent living, improve quality of life, and reduce avoidable health and care costs [7].

As traditional assessment methods are often impractical in routine healthcare [9], it is even more important to create alternatives to reduce the time-consuming evaluation of nutritional data, which is however essential for the prevention and treatment of malnutrition.

Technology-based nutritional assessment tools have the potential to address these issues. When adapted for older adults, they can optimise the data collection process, provide real-time feedback, and reduce the workload of healthcare professionals, offering promising alternatives by simplifying, optimising, and improving the efficiency of food records and evaluation [12].

While many dietary apps exist, few are tailored to the specific needs of seniors.

The NuMob-e-App was co-developed in 2021 with older adults and healthcare professionals to enhance usability and adherence [13]. In an iterative development process, it was further refined until 2022 so that it could be compared against the gold standard in the course of this study [14].

The current version (2.0) of the app allows users to track their meals and physical activities while receiving visual feed-back on nutrient intake and physical performance [14,15]. It was designed to support older adults in (post-)rehabilitation in monitoring and understanding their nutrition and mobility behaviour by offering interactive diaries and personalized feedback.

In particular, it addresses typical barriers in this age group, such as impaired vision, fine motor skills, and limited digital competence [16], and therefore offers visually simplified, intuitive, and low-threshold functions that are easy to use even for people with no technical experience [14].

However, before widespread implementation of a health app, the validity and reliability must be demonstrated through studies.

In the present validation, only the nutritional part of the application was tested.

This study aims to validate the dietary food record function of the NuMob-e-App by evaluating whether it can achieve equivalent results compared to the gold standard method of the 24-hour recall method.

## Materials and methods

### Participants and recruitment

Older people were included in the study if they (1) were at least 70 years old and (2) lived independently in their own home or domestic environment. Exclusion criteria were as follows: (1) Unable to or not willing to give informed consent; (2) insufficient ability to understand the study content and procedure or the German language; (3) patients with cognitive impairment (e.g., dementia) or with dysphagia requiring texture-modified feeding; (4) severe visual limitations that do not allow independent operation of the tablet; (5) simultaneous participation in other studies aimed at influencing dietary behaviour.

Recruitment was conducted via four channels: The Division of Geriatric Medicine at the Department of Health Services Research at Carl von Ossietzky University Oldenburg sent study information to people aged 70 or above, who were generally interested in participating in studies. Furthermore, the junior research group 'Nutrition and Physical Functionality in Older Adults' at the Department of Health Services Research at Carl von Ossietzky University Oldenburg provided a list of individuals who had previously expressed interest in participating in studies and gave informed consent to be contacted from the members of the group; additionally, posters and flyers were distributed in public places frequently visited by seniors (doctor's offices, libraries, pharmacies, senior citizens' meeting places). Snowball sampling was encouraged by providing flyers for participants to share within their social networks.

### Study design

After giving written informed consent, participants first completed a sociodemographic and baseline characteristics questionnaire, in which we using standardized instruments, including the *Mini Nutritional Assessment – Short Form (MNA-SF)* [17], the *Physical Activity Level (PAL)* [18], and the *Technology Commitment Scale* by Neyer [19].

The MNA-SF [17] was used to screen for malnutrition. This validated six-item tool classifies older individuals as 'well nourished', 'at risk of malnutrition' or 'undernourished' and examines aspects such as the person's weight loss, nutritional status and mobility. The *Physical Activity Level* (PAL) [18] was recorded based on participants' self-assigned activity level from a translated version of a standardized classification table. The *Technology Commitment Scale* by Neyer et al. [19] includes twelve Likert-scaled items and has been validated in adults aged 52 years and older to evaluate the digital affinity.

Based on the sociodemographic information collected (e.g., gender, weight, height, age), the app was pre-configured individually for each user to reflect appropriate default values and personalize the feedback features and nutrient requirements. Each participant received a tablet preinstalled with the NuMob-e-App and was individually trained on its use. As part of the training, participants individually documented at least one meal (usually breakfast from that day) together with the study team to become familiar with the app's interface and portion size logic. If desired, they could practice further by

entering additional meals, until they felt familiar with the app's usage. These training entries were not included in the study dataset and served exclusively for practice purposes. All participants received standardized oral and written instructions during the initial training following the informed consent.

Participants were instructed to document all food and beverage intake on three pre-scheduled days which included one weekend and two weekdays, using the app. No specific dietary instructions or restrictions were given, and participants were encouraged to eat and drink as usual. Participants were instructed to take the tablet home for the entire documentation period. They were allowed to integrate it into their everyday lives, for example by taking it to restaurants or social events if they wished. Participants were asked to document their meals during or shortly after eating and permitted to document until midnight of the same day. On each of the pre-scheduled days, a structured 24-hour dietary survey was conducted by telephone.

## The NuMob-e-app

The NuMob-e-App is a digital assistance system refined by the junior research group 'Nutrition and Physical Functionality in Older Adults', based on the digital DiDiER-nutrition diary by Elfert et al [13]. Originally intended for use among older rehabilitation patients, the app was designed to promote sustainable behavioural changes in the areas of nutrition and physical activity by offering tailored, age-appropriate feedback.

The NuMob-e-App offers dual functionality – supporting nutritional and physical activity tracking. For the purpose of this study, however, the physical activity module was deactivated to focus exclusively on the nutrition diary component.

As part of a user-centered design process, several focus groups with older adults were conducted to identify usability barriers, knowledge gaps, and user expectations, especially for this age group, which were then implemented in the user interface [14].

The development of the nutrition modules and the basis for implementing the nutritional recommendations were based on established recommendations for the nutritional needs of older people, including the ESPEN guidelines on clinical nutrition and fluid intake in geriatrics [1] and the information materials of the German Nutrition Society 'Eating and drinking in old age' [20] and 'Malnutrition in old age' [21].

The app enabled users to document their daily food and fluid intake using household-based portion descriptions (e.g., slice of bread, glass of juice, half an apple). Entries are structured by meals (breakfast, lunch, dinner, and snacks).

Based on sociodemographic characteristics such as age, sex, and weight, the app provided automated, personalized feedback and visual summaries of intake compared to the nutritional reference values, including calories, protein, and the seven food groups defined by the German Nutrition Society (germ. 'Deutsche Gesellschaft für Ernährung') (DGE) ('Grain, grain products and potatoes', 'Vegetables and salad', 'Fruit and fruit juice', 'Milk and dairy products', 'Meat, sausage, fish and egg', 'Fats and oils' and 'Beverages') [20].

## The 24-hour-recalls

On each day following app documentation, a standardized 24-hour dietary recall was conducted via telephone by the same study staff member throughout the entire study period, in order to minimize inter-rater variability.

Before starting a recall, participants were explicitly instructed not to refer to their app documentation or other notes and to rely solely on their memory. Each recall followed a consistent protocol based on the five-step US Department of Agriculture automated Multiple-Pass method [22] and was adapted to our needs: (1) Open narrative of the previous day's intake, from morning until night; (2) Clarification of portion sizes and leftovers; (3) Structured non-specific prompting for often-forgotten items (e.g., beverages, snacks, toppings, cooking fats); (4) Summary readback of all recorded entries for participant confirmation. The same study team member conducted all dietary recall interviews, ensuring consistent questioning, documentation, and interpretation throughout the whole study.

This procedure allowed participants to describe their intake freely, but still precisely, including specifying exact weights or preparation details where known. Food items not directly mappable to the nutrient database were harmonized through consultation within the study team and by nutrient equivalence.

Nutrient intake estimates for each 24-hour-recall were served as the reference-gold-standard in the present study.

## Statistical analysis

All nutrition data were processed using the German Nutrient Database (German: Bundeslebensmittelschlüssel; [23]). To ensure data integrity and minimize transcription errors, all entries from the 24-hour recalls and the app documentation were double-entered into REDCap, a secure, web-based platform designed to support data capture for research studies, following a four-eyes principle to ensure data accuracy and integrity. Discrepancies were resolved through cross-checking with the original paper records and app data [24,25]. Deviating entries were resolved by the main reviewer by referring to original handwritten recall- documentation or original app-data.

For statistical analyses, the validated dataset was exported from REDCap to IBM SPSS Statistics (Version 29.0, IBM Corp., NY, USA), and selected summary statistics and graphics were additionally prepared using Microsoft Excel Version 16.96.1) [Microsoft [26], Redmond, Washington, U.S.A.].

To assess the degree of agreement between the NuMob-e-App and the 24-hour-recall method, the following statistical procedures were applied:

(1) Statistical equivalence was tested using the Two One-Sided Tests procedure for the four macronutrients (energy, carbohydrates, protein, and fat) and the seven DGE-defined food groups [20]. Analyses were performed for each of the three documentation days (Day 1 (D1), Day 2 (D2), Day 3 (D3)) and for the averaged values across all three days. A predefined equivalence margin of ±10% was used, based on previous validation work [14], and results were considered statistically equivalent if both one-sided tests yielded p-values below 0.0125 (Bonferroni-corrected).

(2) Agreement was assessed using intraclass correlation coefficients (ICC model 3.1), allowing for comparison across individual and averaged days.

(3) Bland-Altman plots were constructed to visualize mean bias and 95% limits of agreement. These plots visualise the mean differences and upper and lower limits of agreement, calculated as the mean difference ±1.96 SD.

Relative bias values (in percent) were calculated to allow comparison across variables with different measurement units, thereby improving interpretability and comparability.

## Ethical regulations

All subjects provided written informed consent prior to participation and were not financially compensated.

The study was conducted in accordance with the Declaration of Helsinki and registered under DRKS-ID: DRKS00032600.

Ethics approval was obtained from the medical ethics committee of Carl von Ossietzky University Oldenburg (Ethics vote Number 2023−172).

## Results

### Population characteristics

The study was conducted between September 2023 and July 2024 in northwest Germany. A total of 104 older adults aged ≥70 who lived independently in their domestic environment were recruited for the study. Two subjects left the study before the first measurements, one due to lack of technical ability, and one for personal reasons. Additionally, five participants were excluded at different stages of the study due to protocol violations, such as backdating entries or failing to submit any data within the app.

The study flowchart can be seen in Fig 1.

Therefore, a total of 97 older adults (female n = 56 (57.7%); mean age 75.8 ± 4.1 years) were included in the final study. Baseline characteristics and nutritional risk profiles of the subjects are summarised in Table 1.

Nearly one quarter (n = 24; 24.7%) were classified as at risk of malnutrition according to the Mini Nutritional Assessment – Short Form (MNA-SF) and the majority of subjects (n = 81; 83.5%) demonstrated an overall active lifestyle, as indicated by their Physical Activity Level (PAL) values, and reached 42.3 points (SD: 8.25) on the Neyer-Likert-scale.

Data were evaluated at the individual day level (Days 1–3) and as averaged values across all days.

## Equivalence testing

On Day 1, equivalence was confirmed only for protein and the food group 'beverages'. On Day 2, six of the eleven variables reached statistical equivalence: energy, protein, carbohydrates, and the food groups 'grains, grain products and potatoes', 'fats and oils', and 'beverages'. On Day 3, equivalence was again demonstrated for five variables: energy, protein, carbohydrates, 'grains, grain products and potatoes', and 'beverages'.

When intake values were averaged across all three days, equivalence was observed for energy, protein, and carbohydrates, as well as for four of the seven food groups: 'grains, grain products and potatoes', 'milk and dairy products', 'fats and oils', and 'beverages'.

When evaluating results across variables rather than by day, the following could be observed: Only two variables – protein and 'beverages' – met the criteria for equivalence on each of the three days and in the mean. In contrast, the variables fat and the food groups 'vegetables and salad', 'fruit and fruit juice', and 'meat, sausage, fish and eggs' showed consistent divergence from the reference method with no equivalence, regardless of the specific day of assessment.

Energy, carbohydrates, and the food group 'grains, grain products and potatoes' were equivalent on two of three days and in the mean. 'Fats and oils' achieved equivalence on one day and in the mean. Notably, 'milk and dairy products' showed equivalence exclusively in the mean value, while no equivalence was found for any individual day.

## Intraclass Correlation Coefficients (ICC)

Generally, ICCs ranged from 0.677 to 0.968.

The highest agreement was observed for the food group 'grains, grain products and potatoes' in the averaged data (ICC = 0.968; 95% CI: 0.952–0.979), while the lowest was found for fat intake on Day 1 (ICC = 0.677; 95% CI: 0.518–0.781). On Days 2 and 3, as well as in the mean across all days, protein showed the lowest ICC values among all four macronutrients.

Of the four macronutrients, carbohydrates demonstrated the highest ICC values on each of the three individual days, as well as in the mean across days (e.g., Day 1: ICC = 0.850; 95% CI: 0.775–0.899). Furthermore, 'grains, grain products and potatoes' demonstrated the most stable agreement across all days, consistently achieving the highest ICCs among the food groups at each time point as well as in the mean. In contrast, 'fats and oils' showed the weakest agreement on Day 1 and Day 3, as well as in the mean.

## Bland-Altman analysis

First of all, most data points clustered closely around the zero line, with only few individual outliers beyond the upper and lower Level of Agreements.

Macronutrients generally exhibited lower relative bias (+2.35–10.0%), while food groups varied more strongly (–27.3% to +9.7%). Most variables showed a negative bias, reflecting underestimation by the app. Exceptions included the food groups 'vegetables and salad' for all survey days and 'meat, sausage, fish and eggs' for days 1, 2 and the mean, which showed positive bias and were thus overestimated by the app.

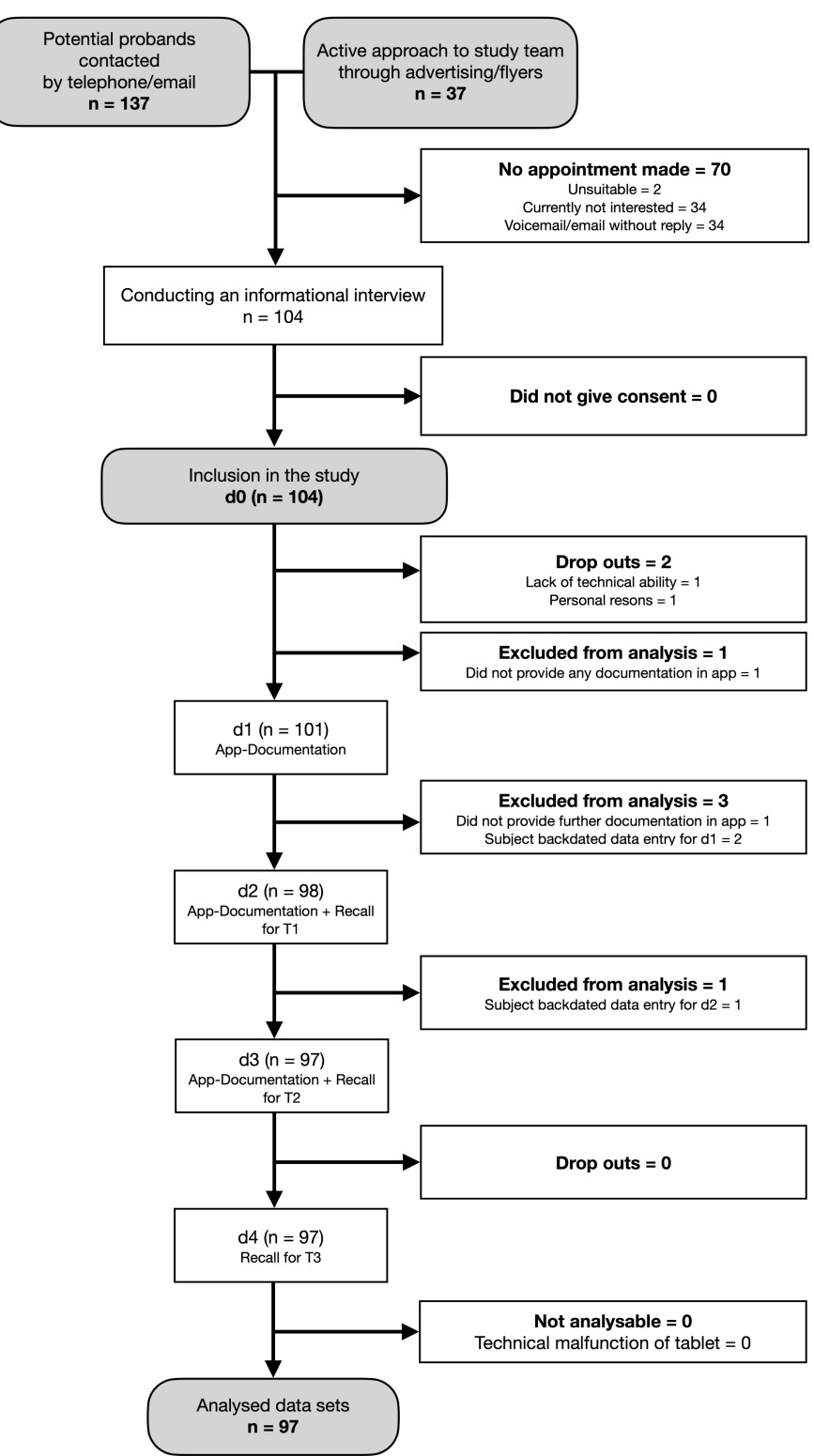

**Fig 1. Subject recruitment and Dropouts.** Note: Dropout: Subjects left study at own request, Exclusion from analysis: Subjects were excluded from the study.

**Table 1. Study characteristics of the subjects in the present study (n = 97).**

| Sociodemographic description | | |
|---|---|---|
| **Total, n** | **97** | |
| Age (years), mean (SD) | 75.8 | 4.13 |
| Sex | | |
| Female, n (%) | 56 | 57.7 |
| Male, n (%) | 41 | 42.3 |
| BMI (kg/m2), mean (SD) | 24.7 | 3.42 |
| Diagnosis | | |
| None of the following, n (%) | 67 | 69.1 |
| Diabetes mellitus, n (%) | 6 | 6.2 |
| Food allergies, n (%) | 14 | 14.4 |
| Gastroenteric disease, n (%) | 14 | 14.4 |
| Kidney disease, n (%) | 3 | 3.1 |
| If yes, dialysis required, n (%) | 0 | 0.0 |
| Medically prescribed diet, n (%) | 0 | 0.0 |
| Medically prescribed drinking restriction | 1* | 1.0 |
| Mini Nutritional Assessment Short-Form | | |
| MNA-SF, mean (SD) | 12 | 1.56 |
| Normal (12–14), n (%) | 73 | 75.3 |
| At risk for undernutrition (8–11), n (%) | 24 | 24.7 |
| Undernourished (0–7), n (%) | 0 | 0.0 |
| Physical Activity Level (PAL) | | |
| PAL-value 1.2, n (%)<br> Only sitting or lying lifestyle | 0 | 0.0 |
| PAL-value 1.4–1.5, n (%)<br> Seated activity with little strenuous leisure activities | 4 | 4.1 |
| PAL-value 1.6–1.7, n (%)<br> Predominantly sitting, also walking/standing activity | 81 | 83.5 |
| PAL-value 1.8–1.9, n (%)<br> Predominantly walking or standing work | 12 | 12.4 |
| PAL-value 2.0–2.4, n (%)<br> Physically demanding occupational work | 0 | 0.0 |
| Technical commitment according to Neyer | | |
| Cumulative score (12–60 points), mean (SD) | 42.3 | 8.25 |
| Females, mean (SD) | 40.04 | 8.05 |
| Males, mean (SD) | 45.46 | 0.35 |

*Note.* SD = Standard deviation, rounded to two decimal places; MNA-SF = Mini Nutritional Assessment – Short Form [27]; PAL = Physical Activity Level [18,28]. Technical commitment was measured using the scale developed by Neyer et al. [19].

*The drinking restriction exactly matched the app's drinking quantity setting of 1,500 ml/day. Therefore, this did not meet an exclusion criterion.

A detailed overview of the relative Bland-Altman results is provided in Fig 2 and the exact Bland-Altman-Plots can be found in the supporting information S1–S11 Figs.

## Discussion

We aimed to validate the dietary food record function of a digital tablet-based e-coach for older adults above the age of 70 years and whether it can achieve equivalent results when compared to the gold standard method of the 24-hour recall.

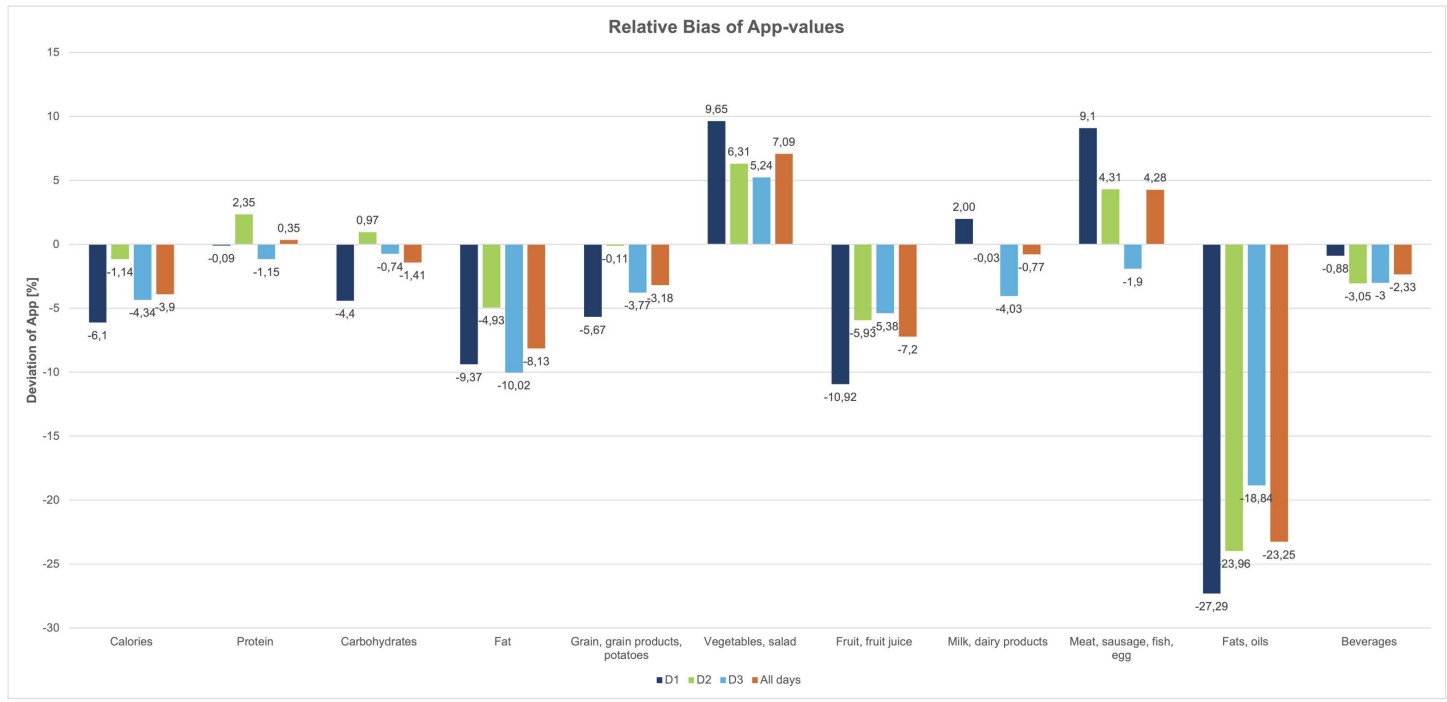

**Fig 2. Relative bias of the App-results.**

We found that: (i) Statistical equivalence (p < 0.0125) was achieved for 20 of the 44 variables tested, including 10 of 16 macronutrient values and 10 of 28 food group values; protein intake and the food group 'beverages' consistently met the equivalence threshold on all three days and on the mean; (ii) intraclass correlation coefficients ranged from 0.677 for fat to 0.951 for carbohydrates, and among the food groups, from 0.714 for 'fats and oils' to 0.968 for 'grains, grain products and potatoes', which consistently showed the strongest agreement; and (iii) Bland-Altman analyses indicated a general pattern of underestimation by the app, except for the food groups 'vegetables and salad' and 'meat, sausage, fish and eggs', which were slightly overestimated. The food group 'fats and oils' exhibited the largest relative underestimation.

## Equivalence testing

Protein and beverages were the most stable variables across all days, while energy and carbohydrate intake reached equivalence from Day 2 onward. Among food groups, only 'beverages' met equivalence consistently. Others such as 'grains, grain products and potatoes' or 'fats and oils' performed variably, and three groups ('vegetables and salad', 'fruit and fruit juice', and 'meat, sausage, fish and eggs') never reached equivalence.

This difference in the observed equivalence rates (macronutrients 10 of 16 variables (62.5%); food groups 10 of 28 variables (35.7%) possibly reflects differences in data aggregation and numerical scale. Macronutrient values are based on cumulative intake and typically involve larger numerical quantities (e.g., calories or grams), which makes them less sensitive to the omission of single food items. In contrast, food groups are recorded in portion units, where small absolute differences can result in large relative deviations. For example, omitting a single portion of 'vegetables and salad' can quickly lead to a deviation of 20% when, for instance, five total portions are expected – enough to exceed the ± 10% equivalence margin and result in a non-equivalence finding.

Furthermore, equivalence was observed in only two variables on Day 1, suggesting a possible learning effect. This pattern aligns with frequently observed improvements in intraclass correlation coefficient (ICC) values and agreement observed on Days 2 and 3.

The 24-hour recall included neutral prompting techniques, while the app did not. This structural difference likely contributed to the finding that statistical equivalence and higher ICC values were more often achieved from Day 2 onward. It is plausible that subjects, after their initial experience with the 24-hour-recall, became more aware of which food components needed to be reported and how detailed their entries should be.

### Intraclass Correlation Coefficients (ICC)

The app achieved high ICCs, ranging between 0.677 and 0.968, particularly for carbohydrates and 'grains, grain products and potatoes', which outperformed other categories across all time points.

The ICC rating framework by Koo and Li [29] classified 19 variables as excellent and 23 as good in terms of agreement.

In comparison, in 2018 Griffiths et. al. [30] assessed the accuracy of five leading nutrition tracking apps by comparing their nutrient intake estimates – entered by researchers based on 24-hour-recalls onto the apps and the 'Nutrition Data System for Research', a dietary analysis software developed for research purposes. They reported correlations ranging from 0.73 to 0.96 for energy and macronutrients, and observed significant underestimations for fat and protein. In our study, fat intake was also clearly underestimated, consistent with these findings. However, protein intake performed notably better: it met statistical equivalence criteria on Days 2 and 3 as well as in the mean, and showed minimal relative bias ranging from −1.15% to +2.35%.

The 24-hour-recall included neutral prompting techniques, while the app did not. This structural difference likely ~~contributed~~ explains why equivalence was often only achieved and ICCs reached better values from Day 2 onward, as subjects likely developed greater awareness of what to report and how they would be questioned. Introducing adaptive prompts in future versions may help close this gap. These prompts could help subjects better anticipate which items are expected and how detailed their entries should be – especially regarding frequently underreported components such as hidden fats or protein-rich foods. Lastly, app design may contribute if prompts for such ingredients are lacking or insufficiently emphasized, making entries less likely unless users are explicitly aware and reminded of them.

### Bland-Altman analysis

Bland-Altman analysis revealed systematic underreporting, especially for fat and the food group 'fats and oils'.

In addition to the already mentioned study by Griffiths et. al. [30] this is also consistent with a study by Chen et al. [31], who evaluated MyFitnessPal entries with two unannounced 24-hour-recalls conducted by researchers using the ASA24 system [32]. Both found that nutrient estimates from consumer nutrition apps were consistently lower than those from the validated reference methods.

Our finding of the general underestimation, particularly for fat and the food group 'fats and oils', may stem from a combination of factors: First, fats are often 'invisible' in meals – such as oils used for cooking or salad dressings – and may be unintentionally omitted due to their less prominent nature [33,34].

Secondly, social desirability bias could discourage full reporting of fat-rich foods, particularly in a health-focused study context. This phenomenon is a common source of underreporting for foods considered unhealthy [33,34].

Conversely, overreporting was observed for 'vegetables and salad' and 'meat, sausage, fish and eggs', potentially linked to social desirability of 'healthy' foods and/or nutrition awareness, which may have led to intentional or unconscious overreporting of protein-rich foods during the study.

### Implications and outlook

The findings of this study suggests that the NuMob-e-App provides a viable and time-efficient alternative to conventional dietary assessment methods such as the 24-hour-recall. While traditional approaches require scheduled interviews and trained personnel, the app allows users to record their food intake independently and in real time, at moments that best fit their daily routines. This not only enhances flexibility and autonomy for older adults, it also reduces the workload for

healthcare professionals. Especially in times of increasing resource constraints in healthcare, digital tools like the NuMob-e-App could contribute to more efficient and practical nutritional monitoring in routine practice.

## Qualities and limitations

This study has several methodological strengths. First, data collection covered three consecutive days, including a weekend day, therefore capturing typical variability in dietary behaviour and enhancing ecological validity [35]. Secondly, the study specifically targeted older adults – a population underrepresented in digital nutrition research – making its findings particularly relevant. Thirdly, in contrast to most existing studies that examined the quality of nutrition apps, this validation study used the same nutritional database (Bundeslebensmittelschlüssel) [23] for both dietary assessment methods, eliminating discrepancies due to differing nutrient sources. Additionally, all data were entered in duplicate using a four-eyes principle to minimize transcription or categorization errors and enhance internal validity.

However, several limitations must also be acknowledged. The study remains subject to typical self-reporting biases, including under- and overreporting due to memory limitations, portion size estimation errors, and social desirability bias [33,34].

The Neyer technical readiness score in our sample was high, but self-selection effects cannot be ruled out. Since participation in the study was voluntary, it is possible that individuals with a higher level of digital literacy and interest were more likely to participate, while those with lower technical skills may not have chosen to participate due to uncertainty or lack of confidence. This could have led to a positively biased sample in terms of digital readiness, which limits the generalizability of our findings to the broader population of older adults, particularly those with low digital literacy.

Additionally, no formal cognitive screening was conducted, so undetected mild impairments cannot be ruled out. This limits the generalizability to cognitively vulnerable older adults, though it also highlights a potential advantage of the app's prospective logging over memory-dependent recall. This also highlights a potential advantage of the app's prospective logging over memory-dependent recall, as it may be feasible even in older adults with minor cognitive difficulties.

Importantly, the findings of this validation study are specific to the NuMob-e-App and cannot be readily generalized to other nutrition tracking apps, as these may differ substantially in terms of user interface, target group, food database, portion estimation tools, and feedback systems.

Lastly, the food group feedback in this study was based on the 2014 version of the DGE guidelines [4]. Since then, the DGE has released updated guidelines [36], merging some categories and introducing new ones (e.g., 'nuts and seeds'). Future updates of the app and validation protocols should align with these changes to ensure continued relevance and comparability.

## Conclusions

This study confirms the relative validity of the NuMob-e-App for assessing the intake of energy, carbohydrates, and protein, as well as selected food groups among independently living older adults. While statistical equivalence was not achieved across all variables, the app performed particularly well for protein and beverages, with excellent to good agreement in most parameters. Bland-Altman analyses revealed a general pattern of underestimation of nutrient intake. These findings support the app's potential for preventive use in self-monitoring and dietary evaluation among older populations, particularly in settings where traditional assessment methods are impractical or financially unfeasible. In view of the constantly evolving field of app development, the present results provide a good basis for future optimisation of the NuMob-e app and comparable tools for the geriatric nutritional assessment.

## Supporting information

**S1 Fig. Bland-Altman-Plots for the difference between App and 24-hour-recall on Days 1, 2, 3 and the mean over the three days for calories.**
(TIF)

**S2 Fig. Bland-Altman-Plots for the difference between App and 24-hour-recall on Days 1, 2, 3 and the mean over the three days for protein.**
(TIF)

**S3 Fig. Bland-Altman-Plots for the difference between App and 24-hour-recall on Days 1, 2, 3 and the mean over the three days for carbohydrates.**
(TIF)

**S4 Fig. Bland-Altman-Plots for the difference between App and 24-hour-recall on Days 1, 2, 3 and the mean over the three days for fat.**
(TIF)

**S5 Fig. Bland-Altman-Plots for the difference between App and 24-hour-recall on Days 1, 2, 3 and the mean over the three days for 'Grain, grain products and potatoes'.**
(TIF)

**S6 Fig. Bland-Altman-Plots for the difference between App and 24-hour-recall on Days 1, 2, 3 and the mean over the three days for 'Vegetables and salad'.**
(TIF)

**S7 Fig. Bland-Altman-Plots for the difference between App and 24-hour-recall on Days 1, 2, 3 and the mean over the three days for 'Fruit and fruit juice'.**
(TIF)

**S8 Fig. Bland-Altman-Plots for the difference between App and 24-hour-recall on Days 1, 2, 3 and the mean over the three days for 'Milk and dairy'.**
(TIF)

**S9 Fig. Bland-Altman-Plots for the difference between App and 24-hour-recall on Days 1, 2, 3 and the mean over the three days for 'Meat, sausage, fish and egg'.**
(TIF)

**S10 Fig. Bland-Altman-Plots for the difference between App and 24-hour-recall on Days 1, 2, 3 and the mean over the three days for 'Fats and oils'.**
(TIF)

**S11 Fig. Bland-Altman-Plots for the difference between App and 24-hour-recall on Days 1, 2, 3 and the mean over the three days for 'Beverages'.**
(TIF)

## Author contributions

**Conceptualization:** Jette Hinrichsen.

**Data curation:** Jette Hinrichsen.

**Formal analysis:** Jette Hinrichsen.

**Funding acquisition:** Rebecca Diekmann.

**Investigation:** Jette Hinrichsen.

**Methodology:** Jette Hinrichsen.

**Project administration:** Jette Hinrichsen.

**Supervision:** Rebecca Diekmann, Vincent Quinten.

**Validation:** Jette Hinrichsen.

**Visualization:** Jette Hinrichsen.

**Writing – original draft:** Jette Hinrichsen.

**Writing – review & editing:** Jette Hinrichsen, Rebecca Diekmann, Vincent Quinten.

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
