## [Decision Letter · Decision Letter 0]

20 Sep 2025

Dear Dr. Hinrichsen,

Thank you for submitting your manuscript to PLOS ONE. After careful consideration, we feel that it has merit but does not fully meet PLOS ONE’s publication criteria as it currently stands. Therefore, we invite you to submit a revised version of the manuscript that addresses the points raised during the review process.

We look forward to receiving your revised manuscript.

Kind regards,

António Raposo

Academic Editor

PLOS ONE

**Journal Requirements:**

1. When submitting your revision, we need you to address these additional requirements. Please ensure that your manuscript meets PLOS ONE's style requirements, including those for file naming. The PLOS ONE style templates can be found at https://journals.plos.org/plosone/s/file?id=wjVg/PLOSOne_formatting_sample_main_body.pdf and https://journals.plos.org/plosone/s/file?id=ba62/PLOSOne_formatting_sample_title_authors_affiliations.pdf 2. Thank you for stating in your Funding Statement: R.D. and V.Q. are supported by the Federal Ministry of Research, Technology and Space (BMBF) (Grant 01GY2102).  Please provide an amended statement that declares *all* the funding or sources of support (whether external or internal to your organization) received during this study, as detailed online in our guide for authors at http://journals.plos.org/plosone/s/submit-now.  Please also include the statement “There was no additional external funding received for this study.” in your updated Funding Statement. Please include your amended Funding Statement within your cover letter. We will change the online submission form on your behalf. 3. We note that you have indicated that there are restrictions to data sharing for this study. For studies involving human research participant data or other sensitive data, we encourage authors to share de-identified or anonymized data. However, when data cannot be publicly shared for ethical reasons, we allow authors to make their data sets available upon request. For information on unacceptable data access restrictions, please see http://journals.plos.org/plosone/s/data-availability#loc-unacceptable-data-access-restrictions.  Before we proceed with your manuscript, please address the following prompts: a) If there are ethical or legal restrictions on sharing a de-identified data set, please explain them in detail (e.g., data contain potentially identifying or sensitive patient information, data are owned by a third-party organization, etc.) and who has imposed them (e.g., a Research Ethics Committee or Institutional Review Board, etc.). Please also provide contact information for a data access committee, ethics committee, or other institutional body to which data requests may be sent. b) If there are no restrictions, please upload the minimal anonymized data set necessary to replicate your study findings to a stable, public repository and provide us with the relevant URLs, DOIs, or accession numbers. Please see http://www.bmj.com/content/340/bmj.c181.long for guidelines on how to de-identify and prepare clinical data for publication. For a list of recommended repositories, please see https://journals.plos.org/plosone/s/recommended-repositories. You also have the option of uploading the data as Supporting Information files, but we would recommend depositing data directly to a data repository if possible. Please update your Data Availability statement in the submission form accordingly. 4. PLOS requires an ORCID iD for the corresponding author in Editorial Manager on papers submitted after December 6th, 2016. Please ensure that you have an ORCID iD and that it is validated in Editorial Manager. To do this, go to ‘Update my Information’ (in the upper left-hand corner of the main menu), and click on the Fetch/Validate link next to the ORCID field. This will take you to the ORCID site and allow you to create a new iD or authenticate a pre-existing iD in Editorial Manager. 5. Your ethics statement should only appear in the Methods section of your manuscript. If your ethics statement is written in any section besides the Methods, please move it to the Methods section and delete it from any other section. Please ensure that your ethics statement is included in your manuscript, as the ethics statement entered into the online submission form will not be published alongside your manuscript. 6. If the reviewer comments include a recommendation to cite specific previously published works, please review and evaluate these publications to determine whether they are relevant and should be cited. There is no requirement to cite these works unless the editor has indicated otherwise. 

Reviewers' comments:

**Comments to the Author**

1. Is the manuscript technically sound, and do the data support the conclusions?

Reviewer #1: Yes

Reviewer #2: Yes

Reviewer #3: Yes

2. Has the statistical analysis been performed appropriately and rigorously?

Reviewer #1: Yes

Reviewer #2: Yes

Reviewer #3: Yes

3. Have the authors made all data underlying the findings in their manuscript fully available?

Reviewer #1: Yes

Reviewer #2: Yes

Reviewer #3: No

4. Is the manuscript presented in an intelligible fashion and written in standard English?

Reviewer #1: No

Reviewer #2: Yes

Reviewer #3: No

**Reviewer #1: ** R1:

General comments:

The author need to add inclusion and exclusion criteria and health state in order to divide study participants into groups to to achieve the study objective, so:

This classification must included in statistical analysis in addition to other subclasses.

o Better to add background

o Good introduction but there is repetition in the idea. author can summarize all introduction in few paragraphs.

o Please re write about the app description and app services.

Methodology:

o Arrange methods as follow:

o Please summarise methods by stratified your methods you do step by step.e, .type of study, study sample, study area, study design, principles of the study ....etc, don't repeat information. It is better to re-write materials and methods.

o Mention the inclusion and exclusion criteria, health status of study participants

o study design is subtitle of methods

o Better to write short description about type of statistical analysis used and its importances, using the scale of significance.

Results:

add descriptive table of study participants to the results

o There is mixing of the three sections(methods, results and discussion)

o Author need to write every section separated from each other including all subtitles, but links to study aim and objective.

Discussion:

1. Give brief introduction about this recent study

2. Discussion of results

3. re-write discussion according to the results.

4. Avoid the numbers that you already mentioned in the result

5. Discuss the finding and compare it with the previous and most resent studies.

6. Defense, confirm and give ethical justification for your results.

7. No subtitles in discussion section

8. Conclusions must be shown and confirmed in the results

o Divide your discussion into sub-title include :

- conclusion

- recommendations

- limitation of the study

**Reviewer #2:**  The manuscript presents a technically sound, original study that meets high standards for scientific research, ethics, and reporting. Its conclusions are appropriately supported by detailed data analysis, and the article is well-written and organized. Data restrictions are reasonably justified by participant consent issues. Only minor improvements in formatting and a deeper discussion on cognitive screening and sample representativeness are warranted. Accordingly, the manuscript is suitable for publication, pending minor revision for presentation and clarity.

**Reviewer #3:**  The subject of this study is interesting. However, there are some points that need consideration and revisions

1. Please make sure to check your in-text citation again. In the third line of the introduction section, your in-text citation is 4??? and after that is 6 and 7??? where is 1-3 and 5 citations???

2. Is there a data related to percentage of elderly people suffering from malnutrition in Germany or any nearby country with similar setting? it is important to include such data to strengthen your idea

3. I recommend the author (s) to seek English editing service

4. The NuMob-e-App, when was its development and release?

5. In table 1, why was smoking not included?

6. The quality of the figure provided is very low. maybe less than 300 dpi??? very difficult to examine it. The supplementary files figures are very clear.

**Do you want your identity to be public for this peer review?** For information about this choice, including consent withdrawal, please see our Privacy Policy

Reviewer #1: **Yes: ** nahla ahmed mohammed

Reviewer #2: No

Reviewer #3: No

---

## [Author Response · Author response to Decision Letter 1]

30 Oct 2025

“Add inclusion and exclusion criteria and health state”

We thank the reviewer for this valuable suggestion. In the revised manuscript, we have added a clear description of the inclusion and exclusion criteria (see Methods section, page 5, lines 77/78).

With regard to the health state, we would like to clarify that only health conditions relevant to the safe conduct of the study (e.g., swallowing difficulties) were assessed, in order to avoid any potential risk for participants. To minimize unnecessary data collection and protect participants’ privacy, we did not record a comprehensive health status, but only information directly required for study participation. Furthermore, a complete assessment of the overall health state was not necessary, as no associations with health of the participants were analysed within the study.

“Good introduction but there is repetition in the idea. author can summarize all introduction in few paragraphs.”

We thank the reviewer for this helpful observation. The Introduction was revised to remove redundancies and is now re-structured. We made some changes in lines 33-35, 45, 51, 60-62 and 69. We believe this improves clarity and readability.

“Please re write about the app description and app services.”

Lines 117-120 and 131-136 describes the app and lines 124-129 describe the development of the app.

“Please summarise methods by stratified your methods you do step by step.e, .type of study, study sample, study area, study design, principles of the study ....etc, don't repeat information. It is better to re-write materials and methods.”

In preparing the manuscript, we carefully followed the section structure and subheadings required by the journal. To ensure consistency with the journal’s format, we prefer to keep the structure of Materials and Methods as following: participants and recruitment, study design, the NuMob-e App, the 24hour Recall, statistical analysis, ethical regulations. Within this structure we did some shortages (line 92, 115 and lines 152/153 and refined the MNA-SF, PAL and technical scale in lines 93-100 with the new requirements.

“study design is subtitle of methods”

In our manuscript, the study design is already presented as a subheading within the Methods section (Level 1 heading = Bold type, 18pt font; Level 2 Heading = Bold type, 16pt font).

“Better to write short description about type of statistical analysis used and its importances, using the scale of significance.”

The statistical methods used in this study (TOST, ICC, Bland-Altman analysis) are already described in the “Methods” > “Statistical analysis” section (lines167-179). Since these are established procedures, we have refrained from further explaining their general principles in order to maintain overview and have instead concentrated on the study-specific application of these methods.

“add descriptive table of study participants to the results”

We agree that participant characteristics are important for context. A descriptive table of study participants is already included as „Table 1: Study characteristics of the subjects in the present study (n = 97)“in the Results section, line 207 ff.

“There is mixing of the three sections(methods, results and discussion)”

After careful review, we did not identify overlaps between the Methods, Results and Discussion sections. Each section is structured according to the journal’s guidelines and presents the required content separately.

“Author need to write every section separated from each other including all subtitles, but links to study aim and objective.”

The sections are already separated and structured according to the journal’s author requirements. To maintain consistency with the journal’s format, we did not restructure the manuscript further.

“Discussion: Give brief introduction about this recent study”

In the discussion section you can find a short summary of the study aim and key findings, which serves as an introduction to the discussion of the results, please see lines 282-283.

“Divide your discussion into sub-title include: conclusion, recommendations, limitation of the study”

In our Discussion, we already present the results structured by statistical methods (Equivalence Testing, Intraclass Correlation Coefficients, Bland–Altman Analysis), followed by Implications and Outlook, and Qualities and Limitations. A separate Conclusion section is also included, in line with the journal’s structure.

“Conclusions must be shown and confirmed in the results”

Please find the conclusion at line 384 ff.

Reviewer#2

“Only minor improvements in formatting and a deeper discussion on cognitive screening and sample representativeness are warranted”

We thank the reviewer for this constructive feedback. We have revised the formatting throughout the manuscript.

Regarding cognitive screening, we clarify that participants were asked whether a diagnosis of dementia had ever been made, which served as an exclusion criterion. However, no specific cognitive test was performed, as already stated in the Discussion, and we emphasized this point further in the revised version (see lines 370-373).

With respect to sample representativeness, we acknowledge that our sample may not fully reflect the general population of older adults, since participants were volunteers in a digital nutrition study and therefore may represent a somewhat more digitally literate subgroup. This limitation was already noted in the Discussion, and we have now elaborated it more explicitly (see Discussion, lines 367-368).

Reviewer#3

“Please make sure to check your in-text citation again. In the third line of the introduction section, your in-text citation is 4??? and after that is 6 and 7??? where is 1-3 and 5 citations???”

We thank the reviewer for this careful observation! The numbering of the in-text citations was indeed inconsistent due to a formatting error with EndNote. We have carefully revised and corrected the citation order throughout the manuscript to ensure that all references now follow the correct sequence (see Introduction, lines 32-34).

“Is there a data related to percentage of elderly people suffering from malnutrition in Germany or any nearby country with similar setting? it is important to include such data to strengthen your idea”

We thank the reviewer for this important suggestion. Indeed, data on malnutrition prevalence among older adults in Germany was reported by Kaiser et al. (2010), showing that 22.8% suffered from malnutrition and nearly every second senior (46.2%) was at risk of malnutrition. We have included this information in the Introduction (see lines 33-35).

“I recommend the author (s) to seek English editing service”

We have carefully revised the manuscript to improve clarity and readability. As the manuscript was already written and reviewed in English at an advanced academic level, we believe that the current version is linguistically sound and does not require additional professional language editing.

“The NuMob-e-App, when was its development and release?”

Information on the development and release of the NuMob-e-App has been added in the revised manuscript (see Introduction, line 59). The App is still part of research and currently does not meet the criteria to be makted as a medical product according to MDR guidelines.

“In table 1, why was smoking not included?”

Smoking behaviour was not assessed in our study, as it was not relevant for the primary aim, which was to compare two dietary assessment methods. Since smoking was not part of the data collection, it cannot be added retrospectively.

“The quality of the figure provided is very low. maybe less than 300 dpi??? very difficult to examine it. The supplementary files figures are very clear.”

The figures have now been revised and re-submitted with improved resolution.

---

## [Decision Letter · Decision Letter 1]

11 Nov 2025

Validation and Evaluation of a Tablet-Based Dietary Record App for Adults Aged 70 and Above

PONE-D-25-42816R1

Dear Dr. Hinrichsen,

We’re pleased to inform you that your manuscript has been judged scientifically suitable for publication and will be formally accepted for publication once it meets all outstanding technical requirements.

Kind regards,

António Raposo

Academic Editor

PLOS ONE

Additional Editor Comments (optional):

Reviewers' comments:

Reviewer's Responses to Questions

**Comments to the Author**

Reviewer #3: All comments have been addressed

2. Is the manuscript technically sound, and do the data support the conclusions?

Reviewer #3: Yes

3. Has the statistical analysis been performed appropriately and rigorously?

Reviewer #3: Yes

4. Have the authors made all data underlying the findings in their manuscript fully available?

Reviewer #3: Yes

5. Is the manuscript presented in an intelligible fashion and written in standard English?

Reviewer #3: Yes

Reviewer #3: The authors have addressed all the comments in professional manners, and included the revisions accordingly.

Their revisions increases the clarity and readability of the manuscript.

I have no additional comments for the authors regarding this manuscript.

**Do you want your identity to be public for this peer review?** For information about this choice, including consent withdrawal, please see our Privacy Policy

Reviewer #3: No

---

## [Editor Report · Acceptance letter]

PONE-D-25-42816R1

PLOS ONE

Dear Dr. Hinrichsen,

I'm pleased to inform you that your manuscript has been deemed suitable for publication in PLOS ONE. Congratulations! Your manuscript is now being handed over to our production team.

Kind regards,

on behalf of

Dr. António Raposo

Academic Editor

PLOS ONE